# Quality Changes and Fungal Microbiota Dynamics in Stored Jujube Fruits: Insights from High-Throughput Sequencing for Food Preservation

**DOI:** 10.3390/foods13101473

**Published:** 2024-05-10

**Authors:** Lili Zhao, Hongbo Li, Zhenbin Liu, Liangbin Hu, Dan Xu, Xiaolin Zhu, Haizhen Mo

**Affiliations:** 1School of Food Science and Engineering, Shaanxi University of Science and Technology, Xi’an 710021, China; zhaolili@sust.edu.cn (L.Z.); zhenbinliu@sust.edu.cn (Z.L.); hulb@sust.edu.cn (L.H.); xudan@sust.edu.cn (D.X.); zhuxiaolin@sust.edu.cn (X.Z.); mohz@sust.edu.cn (H.M.); 2College of Food Science and Engineering, Central South University of Forestry and Technology, Changsha 410004, China

**Keywords:** jujube, storage period, pathogenic bacteria, quality changes, preservation

## Abstract

Postharvest rot is an urgent problem affecting the storage of winter jujube. Therefore, the development of new technologies for efficient and safe preservation is very important. This study aimed to elucidate the fungal microbiota found on the epidermis of jujube during the storage period using high-throughput sequencing, as well as to monitor the changes in quality indexes throughout this period. Through internal transcribed spacer (ITS) sequencing, we identified two phyla (Basidiomycota and Ascomycota) and six genera (Cryptococcus, Bulleromyces, Sporidiobolus, Alternaria, Pseudozyma, and Sporobolomyces), which potentially contribute to the spoilage and deterioration of jujube, referred to as “core fungal taxa”. A high correlation was further found between preservation indices (including decay rate, firmness, and total soluble solids) and the growth of multiple core fungi over time. These findings will provide insights and a theoretical basis for further research on preservation techniques related to biological control during date fruit storage.

## 1. Introduction

The jujube tree, known as Zizyphus jujube Mill., is considered the most important species in the Zizyphus genus for fruit production within the Rhamnaceae family. Originally from China, the winter jujube (Zizyphus jujube. Mill. cv. Dongzao) has been cultivated for 4000 years and is widely available in Europe, southern and eastern Asia, and Australia [1]. Due to its exceptional nutritional content and abundance of functional bioactivities [2], winter jujube fruit is considered a functional food and traditional Chinese medicine. At present, China ranks among the top nations in the world for both jujube fruit production and consumption, with 5.47 million tons produced in 2019 [3].

In recent years, the demand for Chinese jujube, particularly fresh fruit types, has significantly increased in the food industry. However, postharvest pathogenic infections and quality degradation limit the storage life and supply time of fresh jujube fruits [4,5]. Dali winter jujubes are a specialty of Dali County, Weinan City, Shaanxi Province, China. Known for their large, nearly round fruits with smooth and glossy surfaces; thin skins; and milky white, tender, and crispy flesh, they are famously sweet and aromatic. These jujubes contain 19 amino acids required by the human body along with various vitamins, earning them the title “crown of all fruits”. The origin of Dali winter jujubes lies in the loess plateau’s fruit region in northern Shaanxi, Dali County. The area’s suitable climate conditions, fertile soil, rich organic matter, and good moisture retention provide an excellent environment for the growth of Dali winter jujubes. However, these jujubes are currently susceptible to diseases such as rot, black spot, and blister spot [6], posing significant barriers to the development of the jujube industry and resulting in substantial economic losses [7]. Currently, the use of chemicals for preventing postharvest diseases of jujube has been the most effective and common strategy. However, biological control has been recognized as a viable alternative to synthetic chemicals for nearly three decades [8]. Numerous scientists have conducted extensive research on this topic [9,10,11,12].

The fruit microbial community structure refers to the microbial species closely associated with the host and their relative abundance in the community [13]. It has been demonstrated that the structure of microbial communities on the fruit surface and their survival mechanisms play a crucial role in the successful implementation of postharvest biocontrol technology [14]. Saminathan et al. [12] found that variations in the fruit microbiome influence fruit tolerance to biotic and abiotic stresses as well as the quality of the final output. In a separate study, diverse watermelon cultivars with red (PI459074, Congo, and SDRose) and yellow flesh (PI227202, PI435990, and JBush) were analyzed using 16S metagenomics and RNAseq metatranscriptomics, highlighting the connection between host and fruit-associated microbiomes in carbohydrate metabolism during fruit ripening [12]. Kusstatscher et al. identified a significant reduction in microbial diversity (*p* ≤ 0.01) and different indicator species reflecting progressive decay and loss of sugar content in decaying sugar beets by thoroughly assessing temporal changes in the microbiome during storage [10].

In the study of these microbial communities, high-throughput sequencing technologies have been widely adopted. Using high-throughput sequencing of 16SRNA and fungal ITS amplicons, Sui et al. [15] characterized the microbial communities in the peel and flesh tissues of kiwifruit harvested from both open-air cultivation systems and rain-shelter cultivation systems. Their analysis indicated that rain-shelter cultivation reduced the natural disease incidence of kiwifruit, which was partially attributed to the differences in the structure and composition of microbial communities present within and on the kiwifruit. Hou et al. [16] explored how hyperoxic shock could reduce the occurrence of disease and decay in goji berries during cold storage by affecting their inherent resistance and the composition of the fungal community, thereby maintaining their quality characteristics. In a study by Sun et al. [17], ozone water, mancozeb, thiophanate-methyl, and clean water were used as different treatments. They employed high-throughput sequencing to investigate the impact of these different treatments on the structure, composition, and diversity of the phyllosphere microbial community in strawberries. This research is beneficial for the prevention and control of strawberry diseases and provides theoretical support for green production. The application of high-throughput sequencing technologies in the study of fruit microbial communities aids in our understanding of the correlation between fruit microbial communities and the degree of decay.

Previous studies have reported the presence of multiple pathogenic species responsible for jujube fruit deterioration, which may exhibit similar symptoms simultaneously. It leads to confusion among jujube farmers, potentially resulting in ineffective disease control and unforeseen consequences [3]. Inspired by the application of high-throughput sequencing technologies, this study aimed to assess the quality and epidermal fungal flora of jujube from Dali, Shaanxi Province, over a 6-day storage period. The objective was to understand the changes in the fungal microbiota of winter jujube during storage and identify core pathogens causing spoilage, as well as changes in freshness preservation indexes throughout the storage period. This research provides valuable insights into the biological control and extension of jujube storage at different stages.

## 2. Materials and Methods

### 2.1. Fruit and Treatment

The selected fruits were from Dali, Shaanxi Province, and a total of 5 kg of jujubes were purchased. Jujubes of uniform color and without visual defects were selected and randomly assigned into six groups of 30 each. One group of samples was washed with sterile PBS every day and then centrifuged in a sterile centrifuge tube. The resulting supernatant was then decanted and stored at −80 °C for subsequent use in internal transcribed spacer (ITS) sequencing. All samples were collected in an ultra-clean bench environment. Jujubes were photographed daily, and any phenotypic changes, such as mold growth or softening, were recorded. In addition, the quality of the jujube fruit was assessed daily for weight loss, firmness, decay, sugar/acid ratio, total soluble solids (TSS), and VC.

### 2.2. Visual Appearance

Images of the visual appearance of jujubes for each day of the storage period were obtained by means of a camera in a dark room with a single light source. After photographing, the jujubes were returned to room temperature for storage until the end of the experiment.

### 2.3. DNA Extraction, PCR Amplification, and ITS Sequencing

The total DNA from the epidermal washings of jujube collected in Section 2.1 was extracted using the cetyltrimethylammonium bromide (CTAB) method. The quality of the DNA extraction was examined through agarose gel electrophoresis, and the DNA was quantified using a UV spectrophotometer.

The ITS2 region of the eukaryotic (fungi) small-subunit rRNA gene was amplified using slightly modified versions of primers fITS7 (5′-GTGARTCATCGAATCTTTG-3′) and ITS4 (5′-TCCTCCGCTTATTGATATGC-3′). These primers were tagged with a specific barcode per sample and sequencing universal primers at their 5′ ends. PCR amplification was conducted in a 25 µL reaction mixture containing 25 ng of template DNA, 12.5 µL PCR Premix, 2.5 µL of each primer, and PCR-grade water to adjust the volume. The PCR conditions for amplifying the eukaryotic ITS fragments consisted of an initial denaturation at 98 °C for 30 s, followed by 35 cycles of denaturation at 98 °C for 10 s, annealing at 54 °C/52 °C for 30 s, extension at 72 °C for 45 s, and final extension at 72 °C for 10 min. The PCR products were confirmed using 2% agarose gel electrophoresis. Ultrapure water was used as a negative control instead of a sample solution throughout the DNA extraction process to eliminate the possibility of false-positive PCR results. The PCR products were purified using AMPure XT beads (Beckman Coulter Genomics, Danvers, MA, USA) and quantified using Qubit (Invitrogen, Carlsbad, CA, USA). The amplicon libraries’ size and quantity were assessed with an Agilent 2100 Bioanalyzer (Agilent, Santa Clara, CA, USA) and the Library Quantification Kit for Illumina (Kapa Biosciences, Woburn, MA, USA), respectively. A combined library of PhiX Control library (v3) (Illumina) and the amplicon library (expected at 30%) was prepared for sequencing. The libraries were sequenced on 250PE MiSeq runs, and one library was sequenced with both protocols using the standard Illumina sequencing primers, eliminating the need for a third (or fourth) index read.

### 2.4. Assessment of Paralysis and Lifespan

The weight loss rate of jujube was determined using the weighing method. The weight of the jujube was measured every 24 h, and the rate of weight loss was calculated using the following formula [18]:X (%) = (a − b)/a × 100% (1)
where X represents the weight loss rate of dates (%); a (g) represents the weight of dates on 0 days, and b (g) represents the weight of dates after every 24 h.

### 2.5. Changes in Firmness of Jujube

Jujube firmness was assessed using a GY-4 digital fruit firmness tester fruit sclerometer (Qingdao Topco Instrument Co., Qingdao, China) as described in a previous study [13]. The firmness measurement involved pressing the jujube at a distance of 4 mm with a probe diameter of 7 mm. Two measurements were taken at the equator part of each jujube, and the average of these two points was used to determine its firmness. Three duplicates, each with ten fruits, were used to treat each group.

### 2.6. Rate of Decay

Random selection was made from six jujube batches for decay rate analysis [19]. The decay rate of the fruit was investigated every other day employing the subsequent formula:Fruit decay rate (%) = (Rotten fruit number)/(Total fruit number)(2)

### 2.7. Determination of Total Soluble Solids and Sugar/Acid Ratio

Thirty jujube fruits were pitted, and their juice was extracted using a juicer. The total soluble solids (TSS) content, total brix content, and acidity of the jujube juice were measured using refractometers (Zhejiang AIRUP Instrument Co., Zhejiang, China), glucometers (Shenzhen Flow Number Technology Co., Shenzhen, China), and pH–acid meters (Deutu Instruments International Trading (Shanghai) Co., Shanghai, China). The jujube fruit sugar/acid ratio is presented as a result of dividing the total brix content by pH.

### 2.8. Determination of Ascorbic Acid (VC)

The quantification of ascorbic acid (VC) in the samples was performed according to the technique described by Moo-Huchin et al. [20]. The mashed fruit pieces were homogenized (5 g) and extracted for ten minutes using 20 g L^−1^ of oxalic acid. The resulting liquid was then filtered. Subsequently, 10 mL of the filtrate was added to a 100 mL conical flask and titrated with 2,6-dichlorophenol-indophenol until a pink color appeared and remained constant for 15 s. The equation shown below was used to calculate VC:X= [(V_1_ − V_2_) × c × 14]/(m × 5/100) × 100 (3)
where X represents the ascorbic acid content of the sample (mg kg^−1^); V is the volume of 2,6-dichloroindophenol solution consumed for titration (mL); V_0_ is the volume of 2,6-dichloroindophenol solution consumed for the blank titration (mL); T is the titration degree of 2,6-dichloroindophenol solution (mg mL^−1^); A is the dilution multiple; m is the mass of the sample (g).

### 2.9. Statistical Analysis

Each experiment was carried out in triplicate, and the findings were presented as mean ± standard deviation (SE). GraphPad Prism v8.0 software was used for data processing. Statistical significance among groups was determined using one-way analysis of variance (ANOVA) and the LSD test with SPSS software (version 26.0; IBM Corp., Armonk, NY, USA). Visualization of a correlation matrix in R is to use the package corrplot [21].

## 3. Results

### 3.1. Surface Morphology

Visual images of jujubes during 0–6 days of postharvest storage are shown in Figure 1, which can directly reflect the surface morphology and fungal growth status of winter jujube. From the 5th day onwards, it can be clearly seen that the jujubes gradually turn red, the epidermal tissues become dehydrated and shrunken, and obvious microbial hyphae appear on the surface. On the 6th day, the rotting phenomenon of the jujube worsened, and the jujube could not maintain its normal edible form.

### 3.2. Taxonomy and Composition of the Epidermal Fungal Community of Jujube

Core taxa were defined as the OTUs present in all samples. The number of core operational taxonomic units (OTUs) in the date epidermis for days 0–6 were 306, 335, 312, 302, 300, 310, and 310 core OTUs, respectively. A total of 80 core operational taxonomic units (OTUs) were obtained from the floral plot, representing potential core microbiomes of jujube fruit (Figure 2).

The OTUs were classified into five fungal phyla at the phylum level (Figure 3A). The dominant phylum in the jujube epidermal microbiota during storage was Basidiomycota (73.05–90.99%), followed by Ascomycota (6.75–20.37%), Zygomycota (0.01–0.11%), Olpidiomycota (0.01%), and Chytridiomycota (0.01%). Unclassified OTUs accounted for a small percentage (0.60–3.49%). Basidiomycota and Ascomycota were the predominant phyla across all examined samples. Thus, these two phyla may constitute the main components of the fungal microbiome on the jujube phylum epidermis.

At the genus level, 31 different fungal genera were detected in each batch (Figure 3B). The most common genera were Cryptococcus (27.67–50.51%), Bulleromyces (15.95–27.83%), Sporidiobolus (5.36–17.56%), Alternaria (2.19–9.57%), Pseudozyma (1.03–10.39%), Fungi_unclassified (1.56–8.53%), Sporobolomyces (1.53-6.78%), Erythrobasidium (0.85–5.37%), Guehomyces (0.07–7.98%), Eupenicillium (0.04–6.20%), Diaporthe (0.01–4.03%), Pleurotus (0.02–1.63%), Others (1.27–2.60%), and Fungi_unclassified (0.02–1.63%) (Figure 3B). Among these genera, Cryptococcus, Bulleromyces, and Sporobolomyces were present in all six batches and constituted a significant proportion of the samples (Figure 3B).

### 3.3. Changes in the Quality of Jujube

The changes in the quality of jujube fruits during storage are illustrated in Figure 4. Notably, jujube fruits exhibit a high water content, and as the storage period progressed, the rate of loss weight of the fruits gradually decreased due to water loss, rising from 0.35% on 1 d to 1.46% on 6 d (Figure 4A). Over time, the jujube fruits underwent gradual decay, with the decay rate increasing from 0% on 0 d to 73.33% on 6 d (Figure 4C). Concurrently, the internal tissue of the fruits became loose and soft, as indicated by a decrease in firmness from the initial value of 2.06 N to 1.202 N, primarily resulting from the enzymatic breakdown of sugars, protopectin, and other constituents within the jujube fruits. The enzymatic activity led to the breakdown of these substances, causing softening and reduced firmness (Figure 4B).

The sugar/acid ratio is an important indicator of the quality of jujube fruits, which is mainly determined by the ratio of soluble sugar and pH. Figure 4D shows that the sugar/acid ratio briefly increased at the beginning of storage but then decreased and then increased. The physiological ripening and spoilage of jujubes resulted in substantial consumption of soluble solids. Furthermore, it can be seen from Figure 4E that the soluble solids were 20.38% on day 0 and then decreased with an increase in storage time. On the sixth day, the soluble solids were 18.02%. This phenomenon may be attributed to starch hydrolysis within the fruit during storage, followed by an increase in respiratory metabolism, progressively consuming more energy materials. Consequently, the sweetness and soluble solids content of the fruit decreased in the later stages, thereby diminishing its edible value. Jujubes are fruits with high VC content, reaching 24.21 mg kg^−1^ of VC at 0 d. The VC content of jujubes decreased continuously over time to 2.79 mg kg^−1^ at 6 d (Figure 4F).

### 3.4. Correlation between Quality Indicators of Jujube and Fungal Community Structure

The correlation analysis between the changes in the composition of jujube fungi at the phylum level and the alterations in quality indexes during storage is illustrated in Figure 5 and Figure 6.

Changes in decay rate and weight loss were positively correlated with Basidiomycota and Cryptococcus and negatively correlated with Olpidiomycota and Ascomycota in the phylum category (Figure 5). In the genus category, changes in decay rate and weight loss rate were positively correlated with Bionectria, Pseudozyma, and Penicillium and negatively correlated with Erythrobasidium, Sporobolomyces, Pleosporales, and Conlarium (Figure 6). Changes in hardness, sugar/acid ratio, TSS, and VC of date fruits in phylum categories were negatively correlated with Ascomycota, Sporidiobolus, and Bulleromyces and positively correlated with Cryptococcus, Zygomycota, and Basidiomycota (Figure 5). In the genus category, jujube fruit firmness, sugar/acid ratio, TSS, and VC variation were positively correlated with Erythrobasidium, Sporobolomyces, Pleosporales, and Conlarium, negatively correlated with Bionectria, Pseudozyma, Penicillium, and Diaporthe (Figure 6).

## 4. Discussion

Fresh jujube is tasty and rich in nutrients, especially vitamins, minerals, and polyphenols [22]. At the same time, fresh jujubes are also a perishable product with a short postharvest life span, especially fully ripe jujube fruits, which are kept at room temperature for less than a week [23]. Fresh jujubes face enormous challenges during storage due to postharvest diseases caused by various microbial pathogens, which result in huge economic losses every year [23]. For example, black spot rot, a dark brown lesion on the surface of jujube fruit, is caused by Streptomyces sp. [24]. The pathogenic fungus mainly lurks in the jujube fruit before harvest and begins to grow or infest the fruit when the metabolic imbalance and the ability of self-defense mechanisms are drastically reduced due to mechanical damage and the intensification of the aging process of jujube fruits during harvesting or storage. Fruit surfaces have highly diverse and dynamic microbial communities that play an important role in host health and quality [13]. In recent years, there has been interest in the impact of microbial antagonists on the evolution of the fruit microbiome [8].

Therefore, based on the importance of the community on postharvest fruit quality and disease resistance, further studies on microbial community changes are necessary. In this study, the core fungal taxa of the epidermal epidermis of jujube were analyzed by high-throughput sequencing, and the core fungal taxa were two phyla (Basidiomycota and Ascomycota) and six genera (Cryptococcus, Bulleromyces, Sporidiobolus, Alternaria, Pseudozyma, and Sporidiobolus). High-throughput results showed that Basidiomycota was negatively correlated with Ascomycota, suggesting that competition between Basidiomycota and Ascomycota for nutrients and space may be one of their antagonistic mechanisms. The abundance of pathogens in the epidermis of winter jujube remained stable during the storage period, as shown in the microbial diversity in Figure 2 and Figure 3.

The advantages of high-throughput sequencing technology in the detection of fruit epidermal fungi are mainly in its comprehensiveness, high sensitivity, rapidity, and ability to provide abundant data. It is able to detect all microorganisms, including those that are difficult to culture by traditional culture methods, and has high sensitivity to detect microorganisms and identify fungi at very low concentrations. In addition, compared to traditional microbial cultures, high-throughput sequencing can provide a large amount of data in a shorter period of time, and these data can provide detailed information about fungal species, numbers, and possible functions. However, a limitation of high-throughput sequencing is its high cost, especially when large-scale samples are analyzed. At the same time, data processing is complicated by the sheer volume of data generated, which requires sophisticated bioinformatics tools and expertise to process and interpret. In addition, the methods of sample processing and DNA extraction may introduce biases that affect the accuracy of the results. Finally, although the presence of fungi can be detected, knowledge of their specific functions and physiological states is still limited. Therefore, although high-throughput sequencing technology provides a powerful tool in the detection of fruit epidermal fungi, its cost-effectiveness and technical limitations need to be considered when using it [25]. The current limitations of high-throughput sequencing can be effectively attenuated by combining it with other technological tools. For example, combining multiplex sequencing platforms and integrating the data can reduce the bias of a single platform and improve the accuracy and reliability of the results. In addition, combining traditional microbial culture methods can validate and complement high-throughput sequencing results, especially in studies of functional and physiological states. Optimizing the data analysis process using bioinformatics tools and machine learning methods can improve the efficiency and accuracy of data processing. With these strategies, the advantages of high-throughput sequencing in the detection of fruit epidermal fungi can be exploited more comprehensively while overcoming its cost and technical challenges [26]. The accuracy of fruit epidermal fungal detection will also be further improved by combining other technological tools in future studies.

The results of the changes in the preservation indexes of winter jujube showed that the weight loss rate, firmness, TSS, VC, and rotting degree of jujube gradually decreased with the prolongation of storage time (Figure 4A–F), and the surface browning phenomenon was serious (Figure 1). This is consistent with previous studies reported [27]. Weight loss due to fruit water loss is an inevitable physiological process in most fruits, from ripening to senescence. Water loss, possibly caused by evaporation and respiratory processes, leads to a degradation in visual appearance as well as textural quality and juiciness [28]. Reduced firmness is a common textural change and is consistent with previous findings that firmness of ambient-stored jujube fruits tends to decrease with storage time [29]. The main reason for softening is the change in the structure and composition of the cell walls of plant cells. The decrease in firmness of jujube is related to the conversion of water-insoluble pectin into water-soluble pectin [30]. Fruit color is one of the main criteria affecting the appearance and quality of the fruit. Kou et al. [31] found that the postharvest browning mechanism of jujube is as follows: in vitro stress and low temperatures lead to an increase in membrane peroxidation and severe cell deformation, resulting in anaerobic respiration that destroys cellular intervals, leading to enzymes and substrates coming into direct contact with the substrate and causing browning. Contact between enzymes and substrates directly triggers browning.

The results of the heat map in this study showed (Figure 5 and Figure 6) that the core fungal taxa of the epidermis of winter jujube showed a high correlation with the quality of jujube preservation. It is noteworthy that an increase in Basidiomycota, Cryptococcus, Bionectria, Pseudozyma, Penicillium, and Diaporthe leads to an increase in the degree of spoilage and water loss of jujubes, a decrease in hardness, sugar/acid ratio, TSS, and VC content, which are not conducive to the freshness and postharvest storage of jujube and belongs to the category of pathogenic fungi. The quality of jujube caused by Olpidiomycota, Ascomycota, Erythrobasidium, Sporobolomyces, Pleosporales, and Conlarium, on the other hand, can slow down the rotting and deterioration of jujube and may belong to the category of beneficial fungi. Therefore, by targeting pathogenic fungi, cultivating beneficial microbes, and fostering antagonistic interactions during the preservation process, we can extend the shelf life and maintain the freshness of winter jujube. Several studies have also proposed exploring the biological associations between microbial species as a way to determine the source of biocontrol agents. Biocontrol is contributing to the innovation of food preservation technology with its green, safe, and efficient advantages. Studies have shown that in tomatoes, endophytic microbial communities that share ecological niches with lycopersicon pathogens exhibit lower disease incidence [32]. As often exemplified, Fusarium spinosum has been described as having the ability to secrete and accumulate fusaric acid to inhibit the production of 2,4-diacetylresorcinol as an antibiotic by Pseudomonas fluorescens. According to a recent study, D. nepalensis from jujube leaves significantly reduced the fungal degradation of “Dongzao”, with less pathogenic fungi and more helpful bacteria present [33].

In the current research on postharvest preservation of fruits, it was shown that propyl melatonin treatment could delay the senescence of winter jujube and effectively maintain the quality of winter jujube [27]. Iturin A at 512 μg/mL can effectively inhibit the occurrence of soft rot disease in tomato fruits [34]. Appropriate concentrations of melatonin treatment could improve the crispness of date fruits, delay weight loss and hardness decline, and inhibit the changes in the total soluble solids (TSS) and titratable acidity (TA) content of date fruits. The γ-cyclodextrin-cinnamaldehyde inclusion complex (γ-CDCL) has been used to control green mold caused by Penicillium digitatum in citrus [35]. Based on these results, we can combine preservation methods with high-throughput sequencing to screen for the most efficient preservatives and preservation means in the future.

Therefore, we found a correlation between the testing of various quality indicators and microbial community analyses of postharvest winter jujube, which can be targeted to inhibit the pathogenic bacteria identified in the study, increase the abundance of beneficial bacteria, and then develop new technologies for preserving winter jujube. This study provides a theoretical basis and reference for the preservation technology of winter jujube using a new strategy of biocontrol to protect specific quality indicator needs by changing the abundance of these microorganisms and may provide new insights on how to manage these bacteria to optimize the quality of postharvest jujube fruits.

## 5. Conclusions

In this study, we examined the structure and diversity of the fungal community in jujube palm during the storage period. Our analysis of the ITS sequencing results revealed the presence of two phyla (Basidiomycota and Ascomycota), as well as six genera (Cryptococcus, Bulleromyces, Sporidiobolus, Alternaria, Pseudozyma, and Sporobolomyces). The relative abundance and distribution of these major genera varied significantly among individuals. Additionally, we investigated the changes in jujube fruit quality throughout the storage period. In addition, we explored the correlation between core fungal taxa in the epidermis of winter jujubes and jujube preservation quality. It was found that an increase in Basidiomycota, Cryptococcus, Bionectria, Pseudozyma, Penicillium, and Diaporthe leads to an increase in the degree of spoilage and water loss of jujubes, a decrease in hardness, sugar/acid ratio, TSS and VC content, which are not conducive to the freshness and postharvest storage of jujube and belongs to the category of pathogenic fungi. The quality of jujube caused by Olpidiomycota, Ascomycota, Erythrobasidium, Sporobolomyces, Pleosporales, and Conlarium, on the other hand, is able to slow down the rotting and deterioration of jujube and may belong to the category of beneficial fungi.

The change in decay rate and weight loss rate were positively correlated with Basidiomycota and Cryptococcus. Firmness, sugar/acid ratio, TSS, and VC content in jujube fruit were negatively correlated with Ascomycota and Bulleromyces, Sporidiobolus. This gives us the insight that the pathogenic bacteria in the postharvest storage process of jujube can be targeted and controlled to promote the growth of beneficial bacteria and also to form the microbial antagonism of beneficial bacteria to pathogenic bacteria. The results of the study can be used to develop new technologies for preserving winter jujube, to achieve the purpose of extending the shelf life and high-quality utilization of jujube.

## Figures and Tables

**Figure 1 foods-13-01473-f001:**
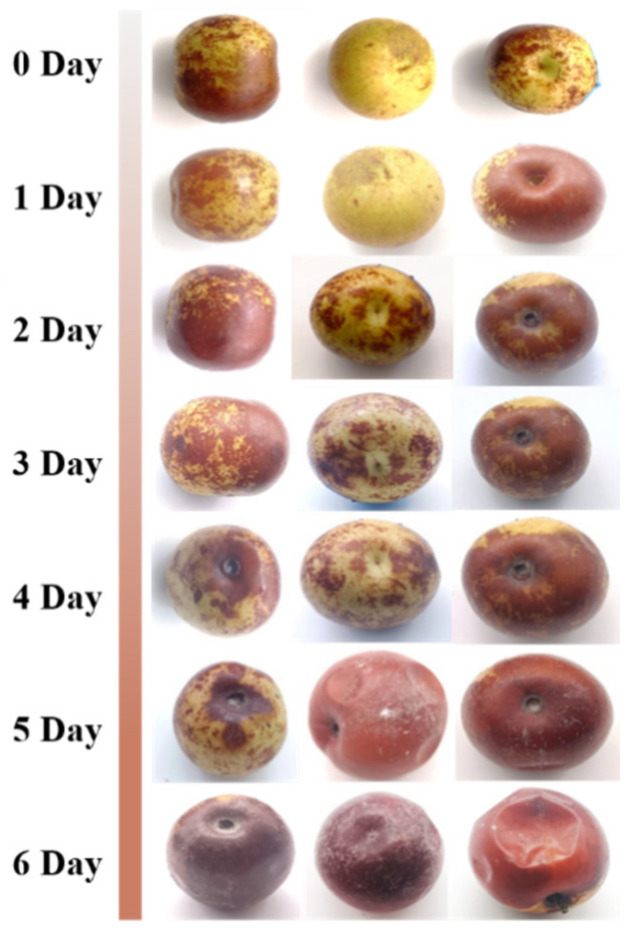
Visual appearance of jujube fruit during 6 days of storage at room temperature.

**Figure 2 foods-13-01473-f002:**
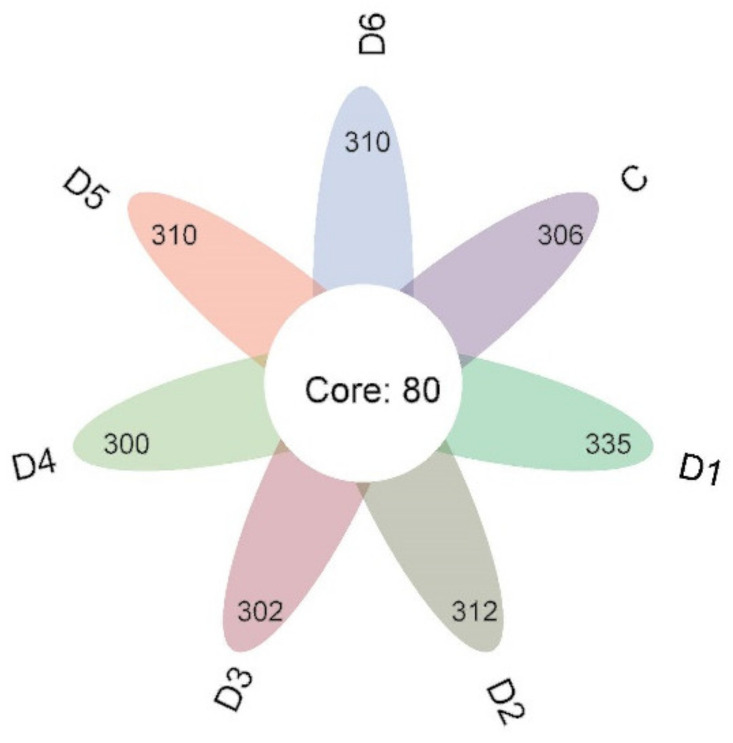
Fungal community composition of jujube fruit samples at the level of genus. The relative abundance of the top 30 in all samples is displayed in detail.

**Figure 3 foods-13-01473-f003:**
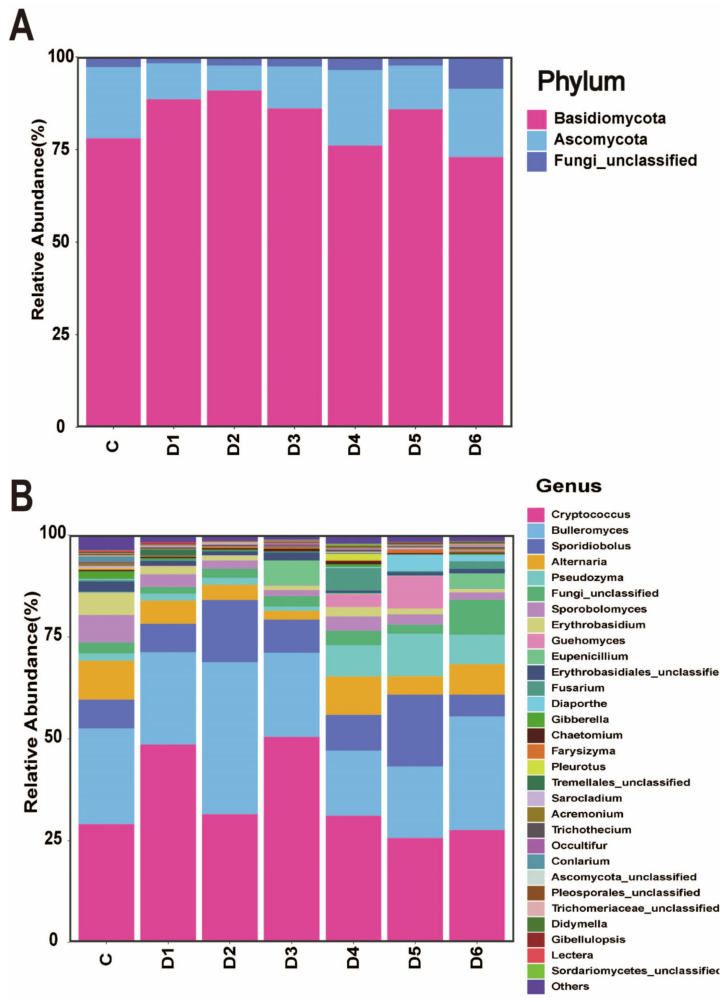
Relative fungal composition of jujube palm epidermis for seven consecutive days. (**A**) Internal transcribed spacer region (ITS) rDNA gene sequences are grouped to the phylum level; (**B**) internal transcribed spacer (ITS) rDNA gene sequences are grouped by genus level.

**Figure 4 foods-13-01473-f004:**
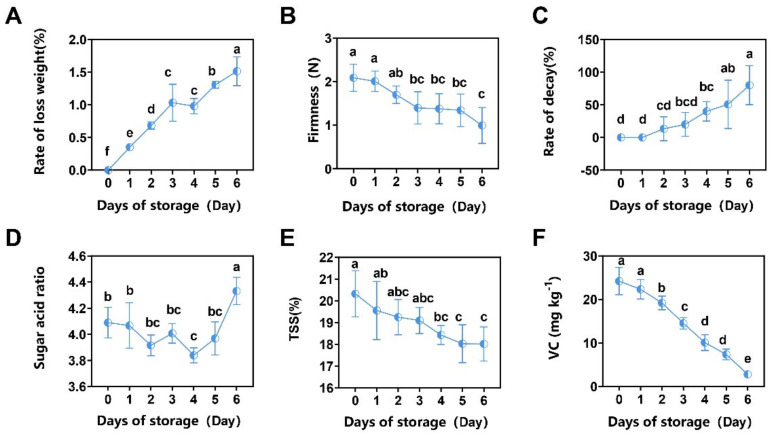
Changes in quality of winter jujube during storage period. (**A**) Changes in jujube weight loss rate; (**B**) changes in jujube firmness; (**C**) changes in jujube of decay; (**D**) changes in sugar acid ratio; (**E**) changes in jujube total soluble solids content (TSS); (**F**) changes in VC (mg kg^−1^). Values with different letter combinations throughout the same storage period showed significant variation (*p < 0.05*, LSD test).

**Figure 5 foods-13-01473-f005:**
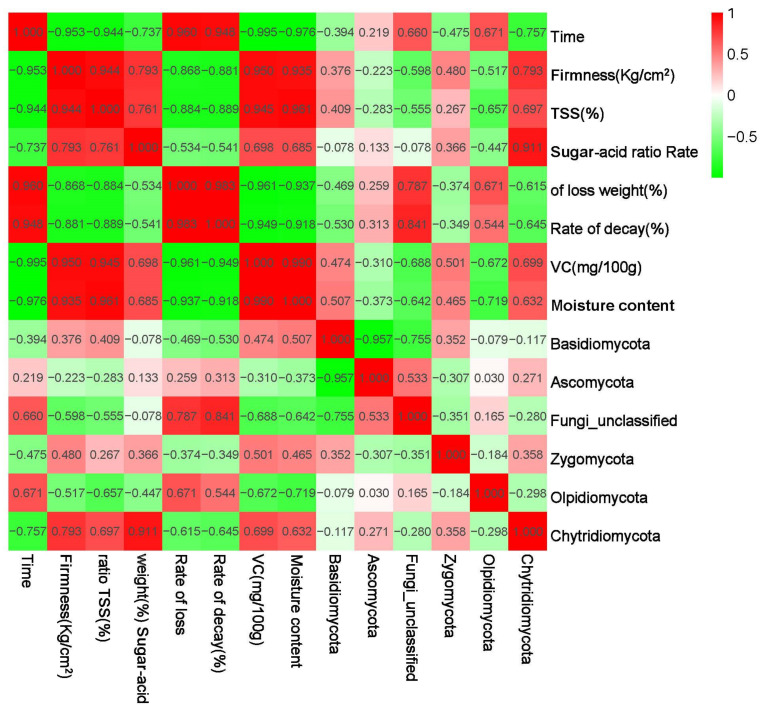
Correlation analysis between compositional changes at the fungal phylum level and changes in quality indexes of winter jujube during the storage period.

**Figure 6 foods-13-01473-f006:**
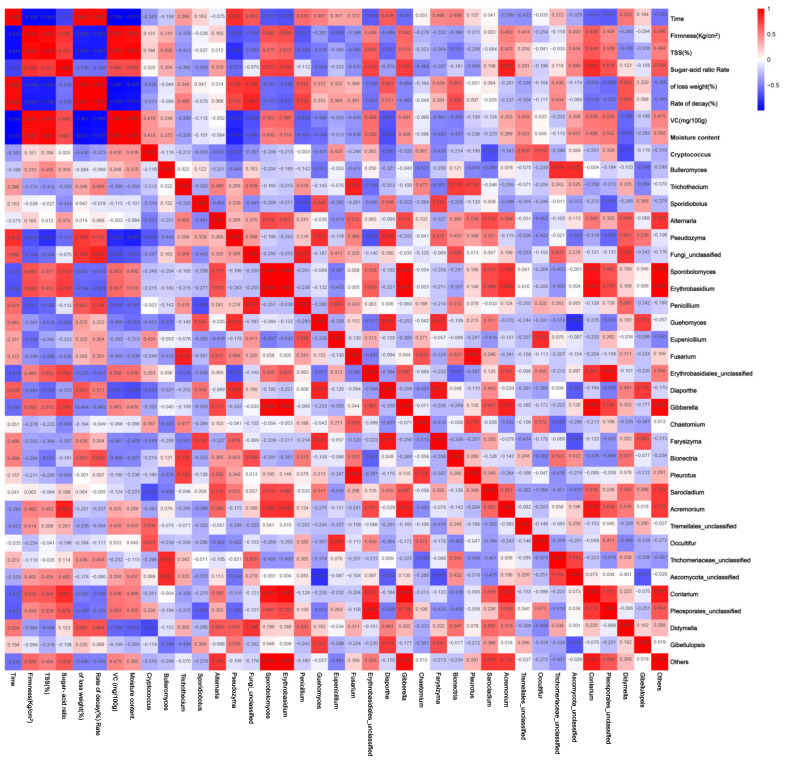
Correlation analysis between compositional changes at the fungal genus level and changes in quality indexes of winter jujube during the storage period.

## Data Availability

The original contributions presented in the study are included in the article, further inquiries can be directed to the corresponding author.

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
