# Peer review of "Quality Changes and Fungal Microbiota Dynamics in Stored Jujube Fruits: Insights from High-Throughput Sequencing for Food Preservation"

_foods, 2024, doi:10.3390/foods13101473_

Round 1

Reviewer 1 Report

Comments and Suggestions for Authors

The article is interesting, and the effort of the authors with the quality of the work is commendable. However, I would recommend some areas for improvement, such as:

Introduction: The article provides a clear exposition of the investigated problem, highlighting the importance of exploring fungal microbiota and changes in the quality of jujube fruits during storage. The introduction adequately establishes the context and justification for the study, although it would benefit from a more detailed description of how high-throughput sequencing techniques have been previously used in similar studies to establish a contrast or advance with respect to those works.

Methodology: The methodology used, including fruit collection, treatments, analysis of fruit quality, DNA extraction, PCR, and ITS sequencing, is described with an adequate level of detail for the replication of the study. Nevertheless, it would be useful to include more information on the criteria for selecting the fruits and justify the choice of a 6-day storage period. Additionally, a deeper discussion on the limitations and advantages of the sequencing methodologies used is recommended. The methodology should contain a detailed description of the specific variety of jujube studied, including information about its genetic characteristics, disease susceptibility, and nutritional quality. Furthermore, the cultivation conditions, such as soil type, irrigation practices, use of fertilizers and pesticides, and crop management techniques, should be detailed. These aspects are fundamental to understanding the context in which the fruits were developed and how these factors can influence the initial composition of the microbiota before storage.

Analysis of Results: The results present valuable data on the dynamics of fungal microbiota and its correlation with changes in the quality of jujube fruits during storage. The identification of fungal taxa and the discussion about their possible role in the preservation or deterioration of the fruits are particularly notable. However, it would be beneficial to expand the statistical analysis to include tests of robustness and sensitivity, which would strengthen the conclusions obtained. Likewise, a comparison with previous studies that have investigated similar fruits or comparable storage conditions would enrich the discussion.

Contributions to Science and Recommendations: The study provides significant insights into the impact of fungal microbiota on food preservation, with a focus on jujube fruits. The implications for the development of biological control strategies are clear and promising. For future research, it would be advisable to explore the effect of different storage conditions and post-harvest treatments on the diversity and dynamics of fungal microbiota.

Comments on the Quality of English Language

Overall, the quality of the English is good. Minor editing of the English language is required.

Reviewer 2 Report

Comments and Suggestions for Authors

Review: Quality Changes and Fungal Microbiota Dynamics in Stored Jujube Fruits: Insights from High-Throughput Sequencing for Food Preservation

The authors examine the microbiome of freshly harvested (postharvest) jujubes over a storage period of 6 days. Therefore, they conducted several quality examinations such as the changes in their firmness, the rate of decay, or their visual appearance. These investigations have been underpinned by sequencing the ITS2 region of various fungal genomes extracted from the microbiome of various jujube fruits.

General remarks:

Chapter 3.2 Taxonomy (Line 168 – 186)

The chapter needs to be completely revised. It has many flaws. You need to introduce a proper definition of your Operational Taxonomic Units (OTU). Let the reader know which organizations are included in your OTU and which are not. In line 175 you highlight two organisms (Stenotrophomonas and Cysticercus), which are predominantly found among your fungal associated groups. Contrary to your experimental section, where your genetic method clearly targets a eukaryotic genome, the Stenotrophomonas is presumably a prokaryote. Moreover, Cysticercus is a tapeworm. I mean, I´m not really an expert when the talk comes to fungal organisms. Additionally, properly organize your taxonomic wording!

Figure 3: Enlarge all the captions

Chapter 3.3 Changes in quality

The chapter needs to be seriously revised, too. You talk about an ANOVA in your materials and methods chapter. Since the data sets presented in that chapter are the only ones highlighted in your manuscript, which are feasible for an ANOVA, I was looking for the results of that statistical evaluation. Are all differences, which can be seen in Fig 4 over the storage period of 6 days, statistically meaningful? Alternatively, does it make more sense to consider the results obtained during the storage as a consistent group (grand mean)?

I don´t follow your description of curve Fig 4E! Generally, when you talk about your curves, mention the values, which also includes the errors, in the text body. Talk about the statistical significance of your measurements.

Chapter 3 is called “results and discussion”. Why did you add an additional chapter called discussion. Talking about chapter 4, please verify every statement you did, if not verified by your own measurements, by literature.

Figure 5: Enlarge all captions

Comments on the Quality of English Language

Although the manuscript is very well written and only needs minor English checks, it contains numerous serious flaws. However, every now and then you´ll find unfinished sentences throughout the manuscript. I recommend to reconsider a publication in “foods” after major revisions.

Round 2

Reviewer 1 Report

Comments and Suggestions for Authors

The authors made all the requested modifications, so the article is suitable for publication. However, I think that the format of the references is not in accordance with the journal's pattern.
